# Sexual Behavior and the Awareness Level of Common Risk Factors for the Development of Cervical, Anogenital and Oropharyngeal Cancer among Women Subjected to HR HPV DNA-Testing

**DOI:** 10.3390/ijerph19159580

**Published:** 2022-08-04

**Authors:** Agnieszka Wencel-Wawrzeńczyk, Piotr Lewitowicz, Angelika Lewandowska, Agnieszka Saługa

**Affiliations:** Collegium Medicum, The Jan Kochanowski University, 25-317 Kielce, Poland

**Keywords:** HPV virus, HPV infection, cancer development, sexual behavior

## Abstract

The human papillomavirus (HPV) is a significant health problem that increases the risk of developing precancerous lesions, cancer of the anogenital area, as well as mouth and throat cancers. The aim of the study was to analyze the awareness level of common risk factors and the preferred sexual behavior of women aged 18–68, who underwent a molecular evaluation of common HR-HPV from material collected from the cervix. The study was conducted at the Jan Kochanowski University in Kielce, Collegium Medicum, in the period from December 2019 to August 2020 on a group of 201 women. A diagnostic survey and the HR-HPV molecular test were used in the research. All of the obtained samples were subjected to identification of and testing for the presence of HR-HPV by the Cobas 4800 platform (Roche Diagnostic^©^, Basel, Switzerland). We presented the statistically significant relationships between the age of the respondents and the awareness of the impact. The women aged over 43 years old presented the widest spectrum of information concerning HPV-related lesions. Conclusion: Our study highlights the necessity of educating women about the prevention of sexually transmitted infections.

## 1. Introduction

The discovery of a cause-and-effect relationship between chronic infection with the human papilloma virus (HPV) and the development of neoplastic changes in the cervix was achieved thanks to the research of Professor Harald zur Hausen from the University of Heidelberg, who, in 1982, for the first time cloned a discussed pathogen [1,2,3]. Careful clinical observations have contributed to the identification of the carcinogenic role of HPV in the formation of cervical cancer, as well as skin lesions, including genital warts. Moreover, expertise on the DNA sequence variation led to the detection of over 200 genotypes of HPV, 85 of which have been precisely verified and 120 have been discussed in a fragmentary way [3,4]. Determining the high level of viral tropism towards the stratified epithelium initiated further analysis of its impact on the development of the head and neck cancers. Additionally, in relation to the characteristics of the head and neck cancers, HPV-dependent changes were observed mainly in the young patients, characterized by a high socio-economic status as well as a good fitness level. Both sexual behavior and preferences in these cases were the main risk factor for the development of an infection. Sexual behavior and preferences are a major risk factor for developing the disease in these cases. It should be emphasized that in recent years there was an increase in the incidence of HPV-related cancers, however, the opposite tendency has been demonstrated in relation to typical head and neck squamous cell carcinomas. Cervical cancer is the most frequently diagnosed malignant tumor in women; its etiology is based on the identification of persistent HPV infection with a high oncogenic potential [5,6,7].

Based on epidemiological data on cervical cancers obtained from 185 countries using the Global Center Observatory (GLOBOCAN) database, Arbyn et al. estimated that in 2018 there were 570,000 new cases and 311,000 deaths caused by cervical cancer. Globally, the average age at diagnosis of cervical cancer is 53 years. Taking into account the global average age of death due to the disease, it was 59 years. Moreover, cervical cancer ranked in the top three cancers in women younger than 45 in 146 (79%) out of the 185 countries assessed [8]. Vaccination against HPV is the basis of prophylaxis in the field of cervical cancer incidence. According to the WHO strategy, by 2020 most of the Member States (55%) have implemented an HPV vaccination program [9].

In Poland, in 2013, this cancer accounted for 3.7% of newly reported cases among women, thus ranking as the seventh leading cause of death due to oncological diseases. Human papillomavirus infection seems to be one of the most common sexually transmitted infections [10]. According to the latest data contained in the Human Papillomavirus and Related Diseases Report, Poland recorded 2137 deaths in 2020. Cervical cancer is the third most common cause of cancer death in women aged 15–44. The annual death toll in Eastern Europe is 15,854 women, and worldwide it is 341,831 women [11].

The tests conducted in order to identify the oncogenic types of HPV were carried out in some of the regions of Poland. One of the studies discussed consisted of collecting material from the cervical canal, in which, using the Amplicor HPV method (Roche Diagnostics), diagnostics were performed in order to diagnose the presence of the oncogenic types of HPV DNA. The study was performed on the basis of the PCR test [12]. It has been estimated that about 80% of sexually active women are infected with HPV in their lifetime, and, at the same time, spontaneous regression occurs in most of the cases. The peak time for acquiring an HPV infection is observed shortly after becoming sexually active and amounts to 40% of the female population under the age of 30. It is worth mentioning that infection usually progresses in a subclinical way. A significant part of the infection disappears after about 2 years. In the case of 5–10% of women, mainly infected with high-oncogenic types, the infection has a persistent form, which is considered the most important etiological factor in the development of cervical cancer [13]. The persistent HPV infection is characterized by the presence of a specific type of virus for a period of at least 6 months or 1 year, however, taking into account the opinion of other authors, it can remain for up to 2 years. Transformation in the form of persistent infection occurs as a result of a reduced cellular immune response and also the impact of the viral oncoprotein E6 and E7 to targeting TP53 and Rb suppressor genes [14].

HPV has been classified into the *Papovaviridae* family. The HPV virion is a small, icosahedral (about 52–55 nm in size) non-enveloped capsid, having a double-stranded circular DNA molecule of 8,000 base pairs. HPV exhibits a strict tropism for the multilayered epithelium of the skin and mucous membranes. A skin or epidermis injury appears to be the means of infection penetrating the basal cell layer. Both the histological form and the clinical features depend on the specific type of HPV DNA located in the area of the pathological outbreaks. It is estimated that, in 8–14% of cases, co-infection occurs with miscellaneous types of the virus. Taking into consideration the features of oncogenic potential, HPV viruses have been divided into such types as: Low Risk HPV types include HPV 6; 11; 40; 42; 43; 44; 54; 61; 72; and 81 as well as High Risk HPV types associated with an increased risk of neoplastic transformation: types 16; 18; 26; 31; 33; 35; 39; 45; 51; 52; 53; 56; 58; 59; 66; 68; 73; and 82 [10,13,14,15,16,17,18].

The issue of human papillomavirus infection (HPV infection) is recognized as a global health problem of the 21st century. Despite the fact that the majority of infections may resolve within 2 years, several well-known genotypes of the pathogen, mainly HPV 16, cause the development of cervical cancers, as well as a significant percentage of other oncological lesions—especially in the anogenital and oropharyngeal areas. Importantly, the oncogenic potential of HPV stems from the cooperative action of the E6 and E7 viral oncoproteins, which disturb the growth-regulatory pathways. The characteristic model of carcinogenesis with HPV viruses assumes that the expansion of the pathogen occurs as a result of viral DNA synthesis into the host genome. The presented clinical picture is observed in 70% of neoplastic lesions within the cervix. In other cases, the tumor cells contain viral DNA in either an episomal form, or both in an episomal and integrated structure. Furthermore, infection occurs most frequently in a latent form, and its source is skin–skin or mucosa–skin contact. It is commonly believed that the main significant factor correlated with HPV infection is sexual intercourse. In the material published in 2019, Baisley et al. observed an increasing trend in HPV incidence rates and the presence of HR genotypes among young girls within the first few years after becoming sexually active. After analyzing their own material, the authors noted a significant association between HPV infection and having more than one sexual partner, as well as the presence of partners in adulthood. It should be emphasized that the differences in the incidence of HPV were strongly related to sexual behavior [16,17,19,20].

In the group of sexual preferences, we can distinguish types of risky behaviors. Additionally, the precise type of conduct depends on the criteria relating to the physical, mental and social sphere. The aspect of body integrity as well as concern for human health and life should be emphasized when considering sexuality from the physical point of view. It should be noted that any sexual behavior aimed at permanent or reversible loss of health, bodily injury or life-threatening activity is considered risky. In spite of the fact that sexual initiation is an important event in the lives of both women and men, it may have different meanings depending on gender. Sexual decision making is influenced by several factors: biological need; curiosity; fascination; gender; age; family conditions; social as well as cultural and religious factors [20,21,22].

It has been confirmed that adolescents very often have doubts about undertaking certain sexual behaviors. In recent years, adolescence has accelerated by at least 4 years. In comparison to other European countries, in Poland, the highest increase in risky behaviors was observed, including the abandonment of contraception and sexually transmitted diseases [22,23]. Recent decades have been marked by progressive changes in sexual behavior among Polish men and women. Over the years, the age of sexual initiation gradually decreased. The following data were observed when analyzing women: 19.34 years in 1997; 19.2 years in 2001 and 18.83 years in 2005. The age of men at the time of first sexual intercourse was presented, respectively: 18.43 years in 1997; 18.32 years in 2001 and 18.06 years in 2005. Taking into account the gender-based differences, on average 27.3% of boys and 18.8% of girls up to 15 years of age, as well as 40.0% of boys and 31.9% of girls in the 17–18 age group, confirm that they have experienced sexual life. Based on the report Sexuality of Polish People, 2002, a decrease in the number of women who declared engaging in sexual contact with one partner was observed [24,25]. Referring to the report from 2020, it was observed that the age of sexual initiation among women ranged between 18–19 years, and among men 17 years [11].

According to some studies, the early initiation of sexual intercourse correlates with a greater number of sexual partners, contraceptive failure as well as premature childbirth. Engaging in early sexual activity can lead to various risky sexual behaviors such as: sexual intercourse with many partners; multiple changes in sexual partners; casual sexual partners; variety of sexual preferences; sexual violence; sponsorship; heterosexual and homosexual prostitution [21,23]. The majority of women who decide to engage in sexual risk behaviors are not aware of the consequences and risks that these decisions bring. Nowadays, despite the fact that HPV is one of the sexually transmitted diseases through vaginal, anal and oral sexual intercourse, the identification of the differences in the choice of sexual behavior in relation to the infection itself is neglected. According to many researchers, early sexual initiation, non-condom use, a high number of lifetime sexual partners, the lack of children and smoking can substantially increase the risk of HPV infection [21,23,26]. Furthermore, the awareness of the human papillomavirus’ transmission is widely acknowledged to be one of the essential components of cervical cancer prevention. Both female education on cervical cancer prevention and the early detection of cervical cancer by medical personnel seem to be unquestionably one of the most important factors in preventing infections with HPV [27].

Today, the literature on persistent infection with oncogenic HPV types confirms the existence of a link between persistent infection with HR-HPV and the formation of cancerous lesions of the cervix, anogenital region, in addition to oropharyngeal tumors [28,29,30].

The aim of the study was to analyze the level of awareness of common risk factors for the development of cancer, knowledge about the HPV virus and the preferred sexual behavior of women aged 18–68 who have undergone molecular evaluation of the common HR-HPV material taken from the cervical canal. Taking into account the high incidence of cervical cancer recorded in Poland, the studies were conducted to assess the impact of sexual behavior and preferences on the aspect of an increased risk of HPV infection.

## 2. Materials and Methods

### 2.1. Organization and Course of the Study

The study was conducted at the Jan Kochanowski University in Kielce, Collegium Medicum, in the period from December 2019 to August 2020. In order to achieve the aims of the study, established research methods and techniques, conditioning the achievement of objective research results, were used. Before starting the study, each patient expressed a willingness and informed consent to participate in the project. Additionally, the diagnostic anonymous survey method, as well as both the survey technique and the HR-HPV molecular test, were also used in the research. The questionnaire study with smears from the cervical canal was carried out in the collection office at Collegium Medicum. The biological material was taken by qualified personnel. This was a prospective study with molecular evaluation of the HR-HPV panel of the material taken from the cervix from students, university staff and women from outside the university. The basic element of recruiting the patients was inviting them to take part in the research and obtaining a written consent to conduct the study. All of the women, after signing a written consent and completing the questionnaire, were tested for the presence of oncogenic types of HPV. Each survey was anonymous, the survey participant was given a unique test code. The participant had the right to decode the test result and generate an official test result capable of further diagnosis or treatment. The study was approved by the Bioethical Commission at Collegium Medicum of Jan Kochanowski University in Kielce, Poland. In all of the cases, informed consent was obtained. The study group consisted of 201 non pregnant women within the age range 18–68 years. Establishing the above-mentioned range was aimed at a precise assessment of preferences and sexual behavior in various age groups.

These groups represent women in the reproductive, perimenopausal and postmenopausal periods. Age was the basic sociodemographic. The technical parameters of the Cobas allow for obtaining positive results of the virus types: HPV-16, HPV-18 and others from the HR-HPV group. A total of 13 patients were positive, of which two were identified with the HPV-16 virus, and 11 with HPV OTHER (as others from the HR group).

A study conducted by our team may contribute to the promotion of activities related to the prevention of the development of HPV-dependent cancers in the area of the mouth and throat, as well as raising the level of awareness regarding the diagnostic possibilities of identifying HPV in the area of the skin and the mucous membranes of the body, as well as the oral mucosa.

### 2.2. Methods, Techniques and Research Tools

The self-designed questionnaire consisted of 17 questions regarding the age of sexual initiation, sexual behavior and preferences, as well as awareness of the impact of HPV infection on the formation of cancer of the anogenital areas and oropharyngeal cancers. The questions from the survey are presented below:Age.Is this your first cervical screening?Have you heard about the influence of HPV on the formation of cervical cancer?Have you heard about the effects of HPV on the formation of oropharyngeal cancer?Have you heard about the influence of HPV on cancer formation in the anogenital area?Are you aware that this test may be performed on men?Are you aware that such test may be taken from the skin and mucosa of the perianal area?Are you aware that such a test can be taken from the oral mucosa?Enter the age of sexual initiation.Enter the number of sexual partners.Whether you use condoms or some other method of contraception?Do you practice oral sex?Do you practice anal sex?Do you think HPV assessment should be funded by NFZ?If there was an option to self-collect a vaginal smear (at home) and send the sample to laboratory, you would be in favor of it?Do you think the way of cervical preventive education is well planned?Where do you get knowledge about the risk of cervical cancer?

The cervical brush-smear was preserved in a dedicated vial provided by the molecular test manufacturer. All of the obtained samples were subjected to testing for and identification of the presence of HR-HPV by the Cobas 4800 platform (Roche Diagnostic^©^, Hong Kong, China).

#### HR-HPV DNA Test Collection Procedure

After hygienic hand washing and disinfection as well as wearing diagnostic gloves, a speculum was inserted into the patient’s vagina and the cervix was visualized. In the next stage, the brush was introduced into the cervical canal. Five rotary movements were performed. The brush was removed and placed in a transport base. By pressing it against the bottom of the container, the biological material was carefully applied.

### 2.3. Methods of Statistical Analysis

The elaboration of the results was based on the statistical analysis of measurable (quantitative) and non-measurable (qualitative) features. The analyses of the relationships between the qualitative variables were conducted with the application of contingency tables, as well as chi-squared tests. The association between these relationships was determined by the Cramér’s V. The statistical analysis was performed with the use of IBM SPSS v26 software (Armonk, New York, NY, USA). A statistical significance of *p* > 0.05 was adopted, indicating the presence of statistically significant relationships or differences.

## 3. Results

Based on the analysis of our own research, it was observed that 33.8% of women under 34 years of age came forward for a cervical examination for the first time in their life. Taking into account the responses of the older women, it was confirmed that 81.5% in the 34–43 age group said that this was not their first study. As presented in Table 1 as a result of the analysis, no significant association was found between the age of the women and the first examination of the cervix (*p* > 0.05).

Based on the study, it was observed that 95.4% of women aged over 43 years of age confirmed that they were aware of the impact of HPV on the formation of cervical cancer. Overall, the vast majority of female respondents, 91.9%, were in favor of having that awareness. In the 34–43 age group, awareness was assessed at 10.4%. Table 2 presents the results of the analysis; no significant association was found between the age of the women and the awareness of the impact of HPV on the development of cervical cancer (*p* > 0.05).

As a result of the analysis, it was found that there is a statistically significant association between the age of the respondents and awareness of the impact of HPV on the development of oropharyngeal cancer, as shown in Table 3. The highest percentage of people aware of this impact was observed in the age group aged over 43 years (68.8%). The second highest result was recorded in the group of people aged under 34 years (54.5%). Meanwhile, the lowest participation of informed people occurred in the 34–43 age group, which amounted to 40.9%. This relationship was supported by tests: χ^2^ = 10.16; df = 2; *p* = 0.006; Vc = 0.228; *p* = 0.006, and its strength measured by Cramér’s V was weak.

Due to our own research, it was found that there is a statistically significant association between the age of the respondents and awareness of the impact of HPV on the development of anogenital region cancer, as shown in Table 4. The highest percentage of people aware of this impact was observed in the age group aged over 43 years (64.1%). The second highest result was recorded in the group of people aged under 34 years (56.9%). Meanwhile, the lowest participation of informed people occurred in the 34–43 age group, which amounted to 34.3%. There was a minimal difference between the total answers of women to yes—101 answers and to no—95 answers. This relationship was supported by tests: χ^2^ = 12.719; df = 2; *p* = 0.002; Vc = 0.225; *p* = 0.002, and its strength measured by Cramér’s V is weak.

The study revealed 53% of the respondents up to 34 years of age and 60% of the respondents aged over 43 years of age, who did not know that HPV testing could also be conducted among men. As a result of the analysis, no statistically significant relationship was found between the age of the respondents and the awareness that the HPV test could also be conducted among men (*p* > 0.05) (Table 5).

The analysis showed the presence of statistically significant relationships between the age of the respondents and the awareness that the test for the presence of HPV can be taken both from the skin and mucosa of the perianal area, as shown in Table 6. The highest percentage of the people aware of this impact was observed in the age group below 34 years (43.9%). The second highest result was recorded in the group of people aged over 43 years (40.6%). Meanwhile, the lowest participation of informed people occurred in the 34–43 age group, which amounted to 20.9%. Based on the survey, more than 79% of the respondents aged 34 to 43 years old were unaware that the HPV test could be taken from the skin and mucosa around the anus. This relationship was supported by tests: χ^2^ = 9.064; df = 2; *p* = 0.011; Vc = 0.215; *p* = 0.011 and its strength measured by Cramér’s V is weak.

As a result of the analysis, it was found that there are statistically significant associations between the age of the respondents and the awareness that the HPV test can be taken from the oral mucosa, as shown in Table 7. The highest percentage of people aware of this impact was observed in the age group below 34 years (44.6%). The second highest result was recorded in the group of people aged over 43 years (37.5%). Meanwhile, the lowest participation of informed people occurred in the 34–43 age group, which amounted to 19.4%. Based on the analysis of our own research, it was observed that about 81% of the surveyed women in the 34–43 age group had no knowledge about the possibility of taking the HPV test from the oral mucosa. This relationship was supported by tests: χ^2^ = 10.013; df = 2; *p* = 0.007; Vc = 0.226; *p* = 0.007, and its strength measured by Cramér’s V is weak.

Based on the conducted research, it was found that there are statistically significant associations between the age of the respondents and the practice of oral sex, as shown in Table 8. The highest percentage of positive responses was observed in the age group aged under 34 (64.1%). The second highest result was recorded in the 34–43 age group (49.3%). Meanwhile, the lowest participation of people who engage in oral sex was in the group of people aged over 43 years of age, which amounted to 32.8%. In connection with the our own conducted research, the surveyed respondents who declared that they engage in oral sex amounted to about 48% of the total, while about 52% do not choose this type of intercourse. This relationship has been supported by tests: χ^2^ = 11.22; df = 2; *p* = 0.004; Vc = 0.239; *p* = 0.004, and its strength measured by Cramér’s V is weak.

Due to our own research, it was observed that over 85% of the respondents do not prefer anal intercourse, while approximately 15% choose this type of sexual contact. As a result of the data analysis, no statistically significant relationship was found between the age of the respondents and the practice of anal sex (*p* > 0.05) (Table 9).

The analysis of our own research allowed us to observe that the majority of the respondents (62.9%) considered that the method of preventive cervical education is not well planned (Table 10).

## 4. Discussion

Nowadays, the dissemination of knowledge about cervical cancer is an important element of the preventive measures aimed at reducing the incidence rate among women around the world. According to the available data, around 80% of cervical cancers are diagnosed in the developing countries. Compared to other European countries, Poland has high rates in terms of diagnosis and also female mortality caused by the disease discussed above. Infection with an oncogenic type of HPV virus is the main etiological factor in the development of disease. According to Błazucka et al. [27], knowledge of the examined women in the field of HPV infections is insufficient.

The analysis of our own research showed that the majority (91.9%) of the respondents confirmed that they had information on the impact of HPV on the formation of cervical cancer, but as many as (45.4%) were not aware of the relationship between the infection and the development of oropharyngeal cancer.

The human papillomavirus (HPV) is the most common sexually transmitted infection. According to WHO, it is responsible for 264,000 deaths worldwide, and requires epidemiological control conducted in accordance with the Global Health Sector Strategy for 2016–2021. We observed that only 5.9% of the respondents had heard and understood the term, HPV. Among them, the most aware people were found in the group aged over 45 years of age [31]. Other analyses indicated that 37% of the respondents declared that they had already heard of HPV, yet only 7% fully understood the impact of viral infection on the development of neoplastic lesions [32].

Based on our own research, it was shown that the highest level of awareness of the impact of HPV on the formation of anogenital cancer as well as mouth and throat cancer occurred in the age group aged over 43 years of age.

Brondani M. et al., proved that 34.5% of the respondents believed that oral sex is an activity with no or low risk for the transmission of HPV [33]. Among the study group, as many as 47.5% of women declared a preference for oral sex, while 14.4% confirmed their engagement in anal intercourse. Referring to the observations of Carlos S. et al. [34] carried out in the Democratic Republic of Congo (DRC), in a group of 718 active heterosexual respondents, 22% had engaged in anal contact, 59% had engaged in oral sex and 18% chose both forms of sexual contact. Based on the study, it was observed that oral and anal sex were common among heterosexual people. Usually, anal practices have been associated with other risky forms of sexual activity. Furthermore, analyzing the views of the respondents from the DRC, oral and anal intercourse are mistakenly perceived as safe, however, in fact, they increase the risk of developing HIV, HPV, HSV, hepatitis B, *gonorrhea*, *vaginal trichomoniasis*, *chlamydia trachomatis* and *treponema pallidum*. The authors of the presented publication also refer to an analysis where, on the basis of material taken from the oral cavity, the following bacteria were detected: *Mycoplasma hominis*; *Mycoplasma genitalium*; *Ureaplasma urealyticum*; *Ureaplasma parvum*, and others that come into direct contact with faeces: hepatitis A; *Salmonella bacteria*; *Shigella*; *Campylobacter* as well as *Entamoeba histolytica*.

Referring to the analysis conducted by Habel M. et al. [35], the data obtained in the years 2011–2015 on the basis of the National Survey of Family Growth questionnaire included a group of 20,621 respondents. The study participants included 11,300 women and 9321 men. The results presented in the above-mentioned publication showed that one third of the respondents confirmed undertaking anal intercourse, of which 11% represented the group of teenagers. The authors of the article observed that two-thirds of the participants aged 15–44 reported engaging in oral sex.

Based on the analysis of our own research, it was shown that the women aged under 34 more often confirmed engaging in oral intercourse (64.1%), while the least frequently discussed practices were observed in the group of the respondents aged over 43 (32.8%). Taking into account the results presented by Sánchez-Vargas L. et al. [36], 72% of the surveyed women systematically undertook oral sexual contact. The average age of the respondents was estimated at 35 years.

Słopieckia A. and Wiraszka G. [37] paid attention to the essence of the insufficient levels of health awareness, which reduces the cure rate of oncogenic diseases. The authors emphasized the importance of promoting health education in order to minimize the risk of developing cancerous diseases of the reproductive organs.

Due to our own research, the majority of the respondents (62.9%) assessed that the method of conducting preventive cervical education is not well planned.

Based on personal observations, a real-time PCR is currently one of the most popular tests for the presence of HPV. It is a method of collecting a sample with a sterile swab from the affected area—in this case, the pharyngeal mucosa. It is worth mentioning that the person undergoing a swab test should retain an empty stomach and should not chew gum, brush teeth or smoke cigarettes 3 h before the procedure. After collection, the material should be properly stored and delivered to the laboratory as soon as possible. Additionally, after analyzing the material using the real-time PCR test, if the result is positive, further diagnostic procedures are carried out and the appropriate treatment is implemented [38]. This study stays in line with previous HPV research in our center, where HPV testing and also p16/Ki-67 testing should play a leading role in screening [39].

## 5. Conclusions

As a result of the current research, it was observed that the highest percentage of the respondents who were aware of the impact of HPV on the development of anogenital cancer occurred in the age group aged over 43. It was presented that the highest level of awareness, regarding the diagnostic possibilities of collecting material both from the skin and mucosa of the anal area as well as the oral mucosa for the presence of HPV, was found in the women aged less than 34 years. The highest percentage of women aware of the impact of HPV on the development of oropharyngeal cancer occurred in the age group aged over 43 (68.8%). Analyzing the sexual preferences of the respondents, the highest percentage of women who engaged in oral sex was recorded in the age group aged under 34 (64.1%), the lowest occurred in the group of the respondents aged over 43 (32.8%).

## Figures and Tables

**Table 1 ijerph-19-09580-t001:** Results from analyzing the relationship between the age of the respondents and the first examination of the cervix.

Age		Is This Your First Cervical Exam?		Total
		Yes	No	
**Up to 34 years**	*N*	22	43	65
	%	33.8%	66.2%	100.0%
**From 34 to 43 years**	*N*	12	53	65
	%	18.5%	81.5%	100.0%
**Over 43 years**	*N*	16	46	62
	%	25.8%	74.2%	100.0%
**Overall**	*N*	50	142	192
	%	26.0%	74.0%	100.0%
**χ^2^ = 3.997; df = 2; *p* = 0.136**				

*N*—group size; %—percentage of the group; χ^2^—Pearson Chi-square test result; df—degrees of freedom; *p*—significance level; Vc— Cramér’s V coefficient.

**Table 2 ijerph-19-09580-t002:** Results from analyzing the relationship between the age of the respondents and the awareness of the impact of HPV on the development of cervical cancer.

Age		Have You Heard about the Impact of HPV on the Development of Cervical Cancer?		Total
		Yes	No	
**Up to 34 years**	*N*	60	6	66
	%	90.9%	9.1%	100.0%
**From 34 to 43 years**	*N*	60	7	67
	%	89.6%	10.4%	100.0%
**Over 43 years**	*N*	62	3	65
	%	95.4%	4.6%	100.0%
**Overall**	*N*	182	16	198
	%	91.9%	8.1%	100.0%
**χ^2^ = 1.782; df = 2; *p* = 0.439**				

*N*—group size; %—percentage of the group; χ^2^—result of the likelihood-ratio chi-square; df—degrees of freedom; *p*—significance level.

**Table 3 ijerph-19-09580-t003:** Results from analyzing the relationship between the age of the respondents and the awareness of the impact of HPV on the development of oropharyngeal cancer.

Age		Have You Heard about the Effects of HPV on the Development of Oropharyngeal Cancer?		Total
		Yes	No	
**Up to 34 years**	*N*	36	30	66
	%	54.5%	45.5%	100.0%
**From 34 to 43 years**	*N*	27	39	66
	%	40.9%	59.1%	100.0%
**Over 43 years**	*N*	44	20	64
	%	68.8%	31.3%	100.0%
**Overall**	*N*	107	89	196
	%	54.6%	45.4%	100.0%
**χ^2^ = 10.16; df = 2; *p* = 0.006; Vc = 0.228; *p* = 0.006**				

*N*—group size; %—percentage of the group; χ^2^—Pearson Chi-square test result; df—degrees of freedom; *p*—significance level; Vc— Cramér’s V coefficient.

**Table 4 ijerph-19-09580-t004:** Results from analyzing the relationship between the age of the respondents and the awareness of the influence of HPV on the development of anogenital region cancer.

Age		Have You Heard about the Influence of HPV on the Development of Anogenital Region Cancer?		Total
		Yes	No	
**Up to 34 years**	*N*	37	28	65
	%	56.9%	43.1%	100.0%
**From 34 to 43 years**	*N*	23	44	67
	%	34.3%	65.7%	100.0%
**Over 43 years**	*N*	41	23	64
	%	64.1%	35.9%	100.0%
**Overall**	*N*	101	95	196
	%	51.5%	48.5%	100.0%
**χ^2^ = 12.719; df = 2; *p* = 0.002; Vc = 0.225; *p* = 0.002**				

*N*—group size; %—percentage of the group; χ^2^—Pearson Chi-square test result; df—degrees of freedom; *p*—significance level; Vc— Cramér’s V coefficient.

**Table 5 ijerph-19-09580-t005:** Results from analyzing the relationship between the age of the respondents and the awareness that HPV test can also be conducted among men.

Age		Are You Aware That Such Test Can Be Conducted among Men?		Total
		Yes	No	
**Up to 34 years**	*N*	31	35	66
	%	47.0%	53.0%	100.0%
**From 34 to 43 years**	*N*	23	42	65
	%	35.4%	64.6%	100.0%
**Over 43 years**	*N*	26	39	65
	%	40.0%	60.0%	100.0%
**Overall**	*N*	80	116	196
	%	40.8%	59.2%	100.0%
**χ^2^ = 1.846; df = 2; *p* = 0.397**				

*N*—group size; %—percentage of the group; χ2—Pearson Chi-square test result; df—degrees of freedom; *p*—significance level.

**Table 6 ijerph-19-09580-t006:** Relationship between the age of the respondents and the awareness that the HPV test can be performed from samples collected from the skin and mucosa of the perianal area.

Age		Are You Aware That Such Test Can Be Collected from the Skin and Mucosa of the Perianal Area?		Total
		Yes	No	
**Up to 34 years**	*N*	29	37	66
	%	43.9%	56.1%	100.0%
**From 34 to 43 years**	*N*	14	53	67
	%	20.9%	79.1%	100.0%
**Over 43 years**	*N*	26	38	64
	%	40.6%	59.4%	100.0%
**Overall**	*N*	69	128	197
	%	35.0%	65.0%	100.0%
**χ^2^ = 9.064; df = 2; *p* = 0.011; Vc = 0.215; *p* = 0.011**				

*N*—group size; %—percentage of the group; χ^2^—Pearson Chi-square test result; df—degrees of freedom; *p*—significance level; Vc—Cramér’s V coefficient.

**Table 7 ijerph-19-09580-t007:** Results from analyzing the relationship between the age of the respondents and the awareness that HPV test can be taken from the oral mucosa.

Age		Are You Aware That Such Test Can Be Taken from the Oral Mucosa?		Total
		Yes	No	
**Up to 34 years**	*N*	29	36	65
	%	44.6%	55.4%	100.0%
**From 34 to 43 years**	*N*	13	54	67
	%	19.4%	80.6%	100.0%
**Over 43 years**	*N*	24	40	64
	%	37.5%	62.5%	100.0%
**Overall**	*N*	66	130	196
	%	33.7%	66.3%	100.0%
**χ^2^ = 10.013; df = 2; *p* = 0.007; Vc = 0.226; *p* = 0.007**				

*N*—group size; %—percentage of the group; χ^2^—Pearson Chi-square test result; df—degrees of freedom; *p*—significance level; Vc—Cramér’s V coefficient.

**Table 8 ijerph-19-09580-t008:** Results from analyzing the relationship between the age of the respondents and the practice of oral sex.

Age		Do You Practice Oral Sex?		Total
		Yes	No	
**Up to 34 years**	*N*	41	25	66
	%	62.1%	37.9%	100.0%
**From 34 to 43 years**	*N*	33	34	67
	%	49.3%	50.7%	100.0%
**Over 43 years**	*N*	21	43	64
	%	32.8%	67.2%	100.0%
**Overall**	*N*	95	102	197
	%	48.2%	51.8%	100.0%
**χ^2^ = 11.22; df = 2; *p* = 0.004; Vc = 0.239; *p* = 0.004**				

*N*—group size; %—percentage of the group; χ^2^—Pearson Chi-square test result; df—degrees of freedom; *p*—significance level; Vc—Cramér’s V coefficient.

**Table 9 ijerph-19-09580-t009:** Results from analyzing the relationship between the age of the respondents and the practice of anal sex.

Age		Do You Practice Anal Sex?		Total
		Yes	No	
**Up to 34 years**	*N*	11	55	66
	%	16.7%	83.3%	100.0%
**From 34 to 43 years**	*N*	9	57	66
	%	13.6%	86.4%	100.0%
**Over 43 years**	*N*	9	54	63
	%	14.3%	85.7%	100.0%
**Overall**	*N*	29	166	195
	%	14.9%	85.1%	100.0%
**χ^2^ = 0.265; df = 2; *p* = 0.876**				

*N*—group size; %—percentage of the group; χ^2^—Pearson Chi-square test result; df—degrees of freedom; *p*—significance level; Vc—Cramér’s V coefficient.

**Table 10 ijerph-19-09580-t010:** Results of the group analysis in terms of the incidence of people who believe that the method of preventive cervical education is well planned.

Do You Think the Way of Cervical Preventive Education Is Well Planned?	*N*	%
**Yes**	73	37.1
**No**	124	62.9
**Overall**	197	100.0

*N*—group size; %—percentage of the group.

## Data Availability

All data derived from this study are presented in the text.

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
