# Peer review of "Sexual Behavior and the Awareness Level of Common Risk Factors for the Development of Cervical, Anogenital and Oropharyngeal Cancer among Women Subjected to HR HPV DNA-Testing"

_ijerph, 2022, doi:10.3390/ijerph19159580_

Round 1

Reviewer 1 Report

Dear Authors

Thank you for your submission of the manuscript.

The study title talks about anogenital cancers however in the results the authors mention oropharyngeal cancers.

The methods lack questions posed to the participants, to allow the reviewer to check for relevancy.

The name of the system used to test the HPV is mentioned however the kit used is not mentioned together with the company. 

it is not clear if the study or recruitment and sample collection was conducted at a clinic, hospital or University.

In the introduction there is repetition of information and a lot of the information provided in the introduction section that is irrelevant to the study.

Results presentation need serious improvement, please use tables and delete most graph presentation they speak of the same results. 

Was age the only sociodemographic data collected by the study?  

Page 8 line 251: Titles of the tables and the questions are misleading for instance table 6 "Results from analyzing the relationship between the age of the respondents and the awareness that the HPV test can be collected from the skin and mucosa of the perianal area." first all the HPV test is a test that is used on samples, the test does not collect samples. The proper way to put this statement would be " Relationship between the age of the respondents and the awareness that the HPV test can be performed from samples collected from the skin and mucosa of the perianal area."

The title of table 8 states "Results from analyzing the relationship between the age of the respondents and the awareness of the impact of HPV on the development of oropharyngeal cancer." However the table itself speaks of practice of oral sex, these two concepts are not the same, there is a serious confusion here. 

page 11 line 305-307, the Authors put a statement that was an instruction to them and an introduction to their discussion "Authors should discuss the results and how they can be interpreted from the perspective of previous studies and of the working hypotheses. The findings and their implications should be discussed in the broadest context possible. Future research directions may also be highlighted.

The discussion needs to be reworded and rewritten focused on the title of the paper and the findings of the paper. 

The conclusion not aligned with the data presented in the results section. 

Please avoid layman language and use scientific language. 

There is about only four questions that deal with risk factors, their presentation in the results is not clear. It is not clear how the questions were asked to the participants, in a e.g questionnaire, Which questions formed part of the study, the information is not clear in the methods.

The whole document needs to be improved. 

Thank you 

Author Response

Honorable Reviewer,

In response to the reviews we have received from you, we have made efforts to improve the content of our manuscript. Thank you for your comments. We present the changes that have been introduced in points. Please see the attachment.

Kind regards,

Agnieszka Wencel-Wawrzeńczyk,

Piotr Lewitowicz,

Agnieszka Saługa,

Angelika Lewandowska

Reviewer 2 Report

The authors assess the association between age and HPV related knowledge and different sexual practices. The authors will need to provide the rational for the study, better describe the recruitment of participants (samples), and better align the objective, methods, and results. More specific suggestions for the authors to consider:  

in the introduction section,

·         clarify the rational to analyze the association between age and HPV related knowledge and different sexual practices

·         explain what makes the women took HR HPV DNA testing different from women who did not take HR HPV DNA testing

·         provide information if there are similar studies done on assessing women’s HPV knowledge and sexual practices in Poland and explain what makes this study different.

In the method section,

·         provide background information of the study site.

·         clarify how the patients were sampled and recruited and the data collection procedures.

·         check and revise statistical analysis methods. Based on the results table, the analysis was to assess the associations between age and HPV related knowledge and sexual practices.  

Author Response

(The authors gave the same response as above.)

Reviewer 3 Report

Comments to the Author:

This study is interesting and would be valuable if the data are correct. The data should be checked again and more accurately presented. The whole Discussion should be re-written.

Abstract:

Page 1, line 10: HPV should be in brackets.

Introduction:

This part is too detailed and must be shortened, particularly the part about HPV, HPV life cycle, cervical cancer disease, sexuality…

The age of sexual initiation is highly questionable herein, since in the rest of the world is deeply under (something above the recommended age for HPV vaccination, 9-15, depending on country).

Materials and Methods:

This part should be divided into chapters.

The age of the study group would be welcomed in this part.

Results:

Some introduction sentences would be desirable herein.

Overall and particularly regarding the question themed in the table 2, possibly different age-span would be better, for example 18-24 and 25-34. Hopefully the difference in these age groups would be more evident.

Page 5, line 201-209: the text would be better to place it after the figure 1. and table 3. Same for the figure 2. and table 4.

The whole text in this chapter should be better composed in regard to figures and tables.

Table 10. should be more clarified.

Discussion:

Whole chapter should be re-written. The Discussion data are undetermined. The first part is redundant.

Author Response

(The authors gave the same response as above.)

Round 2

Reviewer 2 Report

·        

Author Response

(The authors gave the same response as above.)

Reviewer 3 Report

In the title I would avoid “as well as” expression.

Table 10. is not fully translated.

Author Response

(The authors gave the same response as above.)
